# Willingness to Contribute to Bio-Larviciding in the Fight against Malaria: A Contingent Valuation Study among Rice Farmers in Rwanda

**DOI:** 10.3390/ijerph182111575

**Published:** 2021-11-04

**Authors:** Alexis Rulisa, Luuk van Kempen, Leon Mutesa, Emmanuel Hakizimana, Chantal M. Ingabire, Fredrick Kateera, Constantianus J. M. Koenraadt, Michèle van Vugt, Bart van den Borne

**Affiliations:** 1Medical Research Centre Division, Rwanda Biomedical Centre, Kigali 7162, Rwanda; cingabire7@gmail.com; 2Department of Cultural Anthropology and Development Studies, Radboud University, 6525 Nijmegen, The Netherlands; 3Center for Human Genetics, College of Medicine and Health Sciences, University of Rwanda, Kigali 4285, Rwanda; lmutesa@gmail.com; 4Malaria and Other Parasitic Diseases Division, Rwanda Biomedical Center, Kigali 7162, Rwanda; ehakizimana@gmail.com; 5Academic Medical Center, Department of Infectious Diseases, University of Amsterdam, 1012 Amsterdam, The Netherlands; fkateera2011@gmail.com; 6Laboratory of Entomology, Wageningen University & Research, 6708 Wageningen, The Netherlands; sander.koenraadt@wur.nl; 7Center for Tropical Medicine and Travel Medicine, Academic Medical Center, University of Amsterdam, 1012 Amsterdam, The Netherlands; m.vanvugt@amsterdamumc.nl; 8Department of Health Education & Promotion, Maastricht University, 6211 Maastricht, The Netherlands; b.vdborne@maastrichtuniversity.nl

**Keywords:** willingness-to-pay, malaria control, larval source management, rice farming, contingent valuation, Rwanda

## Abstract

There is broad consensus that successful and sustained larval source management (LSM) interventions, including bio-larviciding campaigns, require embeddedness in local community institutions. Ideally, these community structures should also be capable of mobilizing local resources to (co-)finance interventions. To date, farmer cooperatives, especially cooperatives of rice growers whose economic activity facilitates mosquito breeding, have remained under the radar in designing community-based bio-larviciding campaigns. This study explores the potential of rice farmer cooperatives in Bugesera district, Rwanda, to take up the aforementioned roles. To this purpose, we surveyed 320 randomly selected rice farmers who belonged to one of four rice cooperatives in the area and elicited their willingness-to-pay (WTP) for application of Bti, a popular bio-larvicide, in their rice paddies. Results from a (non-incentivized) bidding game procedure, which tested two alternative contribution schemes showed that financial contributions would be significantly different from zero and sufficient to carry a co-financing share of 15–25 per cent. A strong heterogeneity in mean WTP is revealed across cooperatives, in addition to variation among individual farmers, which needs to be anticipated when engaging farmer cooperatives in LSM.

## 1. Introduction

In mid-2020 the government of Rwanda launched a pilot program that used precision drone technology to spray biological larvicides in six high-risk malaria zones. The intervention was mainly targeted at areas of rice cultivation that had witnessed increasing rates of malaria in recent years [1].

Larval source management (LSM) is the latest addition to the country’s malaria prevention package, complementing the distribution of long-lasting insecticidal nets (LLINs) and indoor residual spraying (IRS) campaigns [2]. While it remains to be seen whether aerial spraying by drones is affordable and accessible when scaled up beyond the pilot zones, the WHO stresses the importance of engaging and mobilizing communities for successful and sustainable vector control in general. It specifically states that vector control interventions such as larviciding should be “empowering communities to gain mastery over their risk of disease and ensure sustainable and locally owned development” [3].

The engagement of communities in LSM cannot be taken for granted, however. Recent experiences with embedding LSM in communities, as documented by Mapua et al. [4] in the Morogoro region of southern Tanzania, reveal several challenges on the ground. Apart from supply issues of larvicides, district and ward-level health officials point to low levels of acceptance by the local population, mostly for the perceived risks that bio-larvicides pose to other organisms, but also the lack of adequate funding features prominently in their narratives. It proved not uncommon for larviciding campaigns to be cut short prematurely as funds ran dry. While the acquisition of the larvicide itself was covered, compensation of time for those involved in the actual manual spraying, or in organizational and logistic support, was often not adequately factored into the budget.

Similarly, focus group discussions with villagers in Malawi, which had recent experience with community-based LSM, identified the lack of financial incentives as an obstacle for participation, pointing to inadequate funds for protective gear (e.g., gumboots) [5]. If communities were to hire drones and rent in, or capacitate, drone operators themselves, LSM would likely become an even costlier affair. If such cost would be (fully) borne by the government or international donors, this might eat into the community’s ‘sense of ownership’ of LSM efforts instead. Hence, the mobilization of local monetary resources, if only as co-financing to limited (inter)national funds, appears a crucial pillar on which sustained LSM rests.

Few studies have investigated to what extent the local population is willing to contribute financially to LSM in their area of residence, particularly in the Sub-Sahara African context [6,7,8]. The current study aims to add to this limited body of evidence.

### 1.1. Purpose of the Study

Rather than assessing the potential for community-wide contributions to LSM, we focus on a specific subset of the local population, i.e., rice-farming households. The motivation for this focus is threefold. First, rice farmers are likely among the most affected by malaria. A systematic positive link between rice cultivation and malaria incidence is emerging from empirical work (see Section 1.2), in which irrigated rice paddies are pinpointed as important breeding grounds for mosquito larvae. Hence, laboring in the rice fields increases exposure to mosquito bites, as does the proximity of rice farmers’ homesteads to the fields. This should increase the marginal return on LSM contributions for this group, especially if a lower malaria burden feeds back into higher rice productivity as a result of fewer working days lost.

A second motivation that stems from this link between rice cultivation and malaria concerns a moral imperative to contribute. Their livelihood strategy produces a negative externality on public health and thus represents the alleged source of an increased disease burden affecting the wider community. Provided that they are aware of their prominent role in local malaria transmission, they may feel a particular responsibility to engage in LSM. This also depends on the extent to which the impact of rice cultivation on malaria is acknowledged more generally in the community. Higher awareness could result in contributions to LSM becoming part of a ‘social license to operate’ for rice farmers.

A final motivation is more pragmatic and relates to the social organization of rice farmers. Farmers tend to operate in membership-based cooperatives and, as such, routinely contribute to collective endeavors, such as the collective purchase of agricultural inputs or the set-up of a solidarity fund for emergencies. Such contributions are often discounted from the pay-out of cooperative sales revenues to individual farmers, so that feeding an LSM contribution into this routine seems more feasible than setting up a community-wide contribution scheme for this specific purpose.

The main goal of the study is to assess whether rice-farming households are willing to contribute financially, and if so, how substantive these contributions would be in view of the actual cost of (low-tech) LSM application. To this purpose, 320 rice farmers in Bugesera district of southern Rwanda were recruited to participate in a WTP bidding game for a manual spraying campaign of *Bacillus thuringiensis* subsp. *israelensis* (Bti) in their fields. WTP elicitation took place in an early stage of an international research project on community-driven malaria control, at which point in time (2015) rice farmers had no prior exposure to larviciding initiatives in the area, but community discussions about suitable local malaria control measures had been facilitated by the project, so that they may have perceived the idea of a larviciding pilot as a real prospect.

We zoom in on WTP variation across individual farmers, based on demographic, socioeconomic and attitudinal characteristics, but also take an interest in variation across cooperatives. The sampled farmers represent four different local rice cooperatives in the sub-district of Ruhuha. Given the potential of cooperative structures as entry point for mobilizing LSM resources, we built two alternative contribution schemes, familiar to cooperative members, into the bidding game. The two options concern a scheme in which farmers contribute equally, regardless of individual landholdings (lump-sum scheme), and a progressive scheme where a fixed contribution rate is set per unit of land owned. We study the relative performance of these schemes across cooperatives. The composition of a farmer’s group, as well as the solidarity norms that have developed within such groups, may shape the relative feasibility and desirability of these alternatives. More generally, we anticipate a complex interplay of personal, household and cooperative-specific factors to shape WTP.

### 1.2. Linking Malaria and Rice Cultivation

From early studies into the effect of rice cultivation on malaria incidence, spanning the 1990s and early 2000s, the so-called ‘paddies paradox’ emerged [9]. Counterintuitively, rather than pushing up local malaria infection risk, malaria incidence in rice areas was typically found to be on par with, or even lower than, incidence in comparable non-rice areas. The main explanation put forward for this paradox concerns a positive wealth effect from rice production, which would effectively counterbalance the impact of increased mosquito density, such as via upgraded housing. However, a recent meta-analysis reveals that while the paradox held up in Sub-Saharan Africa until the early 2000s, this seems no longer the case [10].

Later studies tend to find consistently that rice cultivation increases the local malaria burden. The authors explain that conceived wisdom thus needs revision and reconcile the contradictory results from earlier and later studies by pointing out that a general reduction in malaria transmission rates has been achieved over time, implying that the context in which the rice-malaria link operates, has changed significantly. The systematic review pools results from 53 studies to estimate that malaria transmission is twice as high in rice communities than in non-rice ones. We have indirect evidence that our research site in Bugesera district fits this picture.

At the time of research, Bugesera district featured among the 19 districts, out of a total of 30, that were classified as carrying a high malaria burden [11]. It is also one of the country’s main rice production areas. While rice is produced in marshlands and low-lying valleys across the country, promotion of rice in Bugesera by the government was particularly strong, as Bugesera historically faces high levels of food insecurity, which local rice production should redress.

A first piece of evidence that links irrigated rice cultivation to malaria exposure stems from a cross-sectional survey among 520 households in Ruhuha sub-district, which is the focal area of this study, in the period April–October 2011 [12]. Households were not sampled randomly, but traced from health center visits. The survey aimed to follow up on 769 patients who reported fever at the outpatient facility located in the area at the start of the mentioned period. A number of 520 patients were effectively traced and completed the survey in their home. Also, all members in these households were screened for malaria through a rapid diagnostic test, covering 2634 individuals, including adults and children. Close to one fifth of the households revealed at least one case of malaria infection and the overall malaria prevalence rate (at individual level) was five percent. When the authors analyzed the degree of spatial clustering of these positive cases, two clusters arose. While the link with rice farming was not established directly, the researchers stressed the fact that the bigger of the two clusters, with a radius of 5 km, neighbors marshlands where traditional rice cultivation is done.

This same cluster, which overlaps with one of the thirty-five villages that make up Ruhuha sub-district, i.e., Gikundamvura, is also pinpointed as a concentration point of malaria transmission in a more comprehensive study carried out two years later (June–November 2013). This result stems from the baseline survey of a broader multidisciplinary research project, which covered 3968 households out of an estimated total of 5100. Again, the survey was complemented with rapid diagnostic tests of all household members (*n* = 12,965). The overall malaria prevalence rate proved stable at five per cent, affecting 13 per cent of households studied. It was calculated that living in Gikundamvura was associated with a significantly increased malaria risk relative to other villages. The authors cautiously state that “Gikundamvura is an area surrounded in the northeast by a vast expanse of marshland used for rice cultivation and probably the marshlands support mosquito breeding and increased malaria transmission risk for neighboring households” [13].

### 1.3. Potential of Biological Larviciding

At the time malaria is declining in most African countries, new strategies targeting mosquito larvae are becoming necessary to be integrated in the set of current vector control interventions that features long-lasting impregnated nets (LLINs) and indoor residual spraying (IRS), both targeting adult mosquitoes indoors. The advantage of larval source management (LSM) is twofold, as it reduces the quantity of mosquitoes biting outside as well as those entering inside the dwellings. It addresses malaria transmission that escapes LLIN, and LSM is therefore strongly complementary.

There are three main types of LSM tools; environmental, chemical and biological. Environmental methods prevent the creation of mosquito breeding sites and include, among others, intermittent rice irrigation and habitat modification. Some success has been reported. For example, Peru has managed to bring down the mosquito larval density, thanks to intermittent rice irrigation for malaria control that creates unfavorable conditions for mosquito reproduction [14]. Water management has also been successful in reducing larval mosquito habitats in Niger, by planting and harvesting as well as carefully draining lands that were lying fallow [15].

A second LSM type is chemical in nature and uses different forms of insecticide or silicone-based surface films. In Kenya, its application in rice fields effectively prevented the emergence of adult mosquitoes by reducing densities of aquatic larval stages. No side effect on abundance of non-target animals and human beings, or growth and development of rice plants has been reported [16].

The third type of LSM, with which we are concerned in this paper, is biological control by making use of natural enemies of targeted mosquitoes and by applying biological toxins. Review studies on malaria vector control strategies document applications across a host of malaria-prone habitats in Asia and Africa [17,18]. A relatively long-standing practice is the use of predators in the form of larvivorous fish, which has proven effective in rice fields in China and India, but often comes at the cost of seriously disrupting existing ecosystems [19]. The record of biological toxins contrasts favorably, also to chemical alternatives. The efficacy of Bti has proven to be high, both in terms of the control of mosquito larvae as well as in killing adult mosquitos, as long as breeding sites are easily identifiable and well-delineated [20,21]. A study in Burkina Faso shows that even at a very low dosage of 0.2 kg/ha, Bti is highly efficacious in reducing malaria mosquito larvae [22]. In Taiwan, Bti was successful in reducing mosquito larvae in rice paddies [23]. Apart from its track record as effective mosquito control agent, its biological nature ensures that any negative impact on non-target organisms is very limited [24]. Tests in the laboratory but also in rice fields conclude that Bti is safe for fish, birds and mammals, including livestock and pets. In fact, most invertebrate species cannot be killed by Bti [25].

Perhaps more important from a farmer’s perspective, rice plants also proved unaffected and their growth was not hampered by Bti application in a Kenyan trial [16]. Furthermore, no occurrence of resistance to Bti has been detected in studies till date. A final merit of Bti concerns the cost aspect. Larval source management tends to be less costly than IRS and LLINs, especially in areas with moderate and focal malaria transmission where mosquito breeding sites are easily identified and well defined [13,26].

A recent systematic review [27] confirms the effectiveness and feasibility of biological larvicide interventions across Sub-Saharan Africa, although it is stressed that such interventions need to be persistent and well-adapted to the behavioral pattern of the local malaria vector. An effectiveness study of Bti in Ethiopia and Kenya also reveals substantial site-specific variation in effectiveness, suggesting optimal success in low-transmission sites where larviciding is supported by high LLIN use as well as community education and mobilization [28].

Currently, LLINs and IRS are being rolled out on a large scale throughout Sub-Sahara Africa, but larval source management lags behind. This contrasts with developments in other regions, where LSM has already found its way into the package of mosquito control measures [29]. Provided that LSM is thoroughly implemented, which implies sufficient coverage of mosquito larvae habitats, there is a long-standing consensus that it is a promising policy option for African countries [14]. Biological larviciding stands out, but like LSM in general, the success of Bti interventions hinges crucially on reliable sources of funding [27,30]. While the cost of Bti is moderate compared to other control methods, it is far from negligible in low-income settings.

### 1.4. Willingness to Contribute to Larviciding

Our WTP study builds on a variety of prior studies, yet the application of a bidding game procedure to elicit rice farmers’ willingness to contribute financial resources to biological larviciding is, to our best knowledge, unique. The bidding game, as an iterative technique of contingent valuation, has been the most frequently used method to assess WTP for malaria control interventions according to a meta-review covering 24 studies up to 2012 [31]. Its authors are able to extract a mean WTP estimate, including zero values for non-contributors, of US$2.79 for insecticide-treated nets, and US$2.60 for “other preventive services” across these studies. The latter figure is difficult to interpret, however, as it includes interventions as diverse as IRS, test kits, and hypothetical vaccines.

No WTP assessments of biological larviciding were identified, but one of the studies covered by the review presents chemical larviciding to bidding game respondents in Sudan [6]. This Sudanese study provides a WTP assessment for (ex-ante) chemical larviciding, as compared to three other vector control methods, i.e., IRS, outdoor spraying, and insecticide-treated nets, among a random sample of 720 households in the states of Gezira and Khartoum. Forty-five per cent of the households, representing a mix of urban, peri-urban, and rural residents, stated a positive willingness-to-pay for chemical larviciding. This willingness increased in household’s economic status, but less markedly so than for IRS and bednets. By contrast, WTP for chemical larviciding responded relatively strongly (and positively) to education levels [6].

Specifically, on WTP for bio-larviciding, we have only come across two studies to build upon. Diiro et al. [8] assess WTP for a novel Bti-formula (UZIMAX) in coastal Kenya (Malindi county). Instead of a contingent valuation method, they use a Becker–DeGroot–Marschak (BDM) auction method to reveal WTP for this product. BDM represents a strong WTP elicitation design, as it is an incentivized auction, although it should be noted that participants do not pay out of their pocket but receive an endowment from the researchers that can be spent (or kept) during the procedure depending on valuation of the product being auctioned off. More than half of the 204 Kenyan participants were from rural communities (57 per cent), while the remainder represented urban dwellers.

The mean WTP amounted to US$4.20 for a specific quantity (250 mL) of larvicide, which the authors calculate should suffice to collectively cover a full two-month rainy season. A participant household’s per capita income significantly increased WTP bids, as did household size, male gender, and having a family member sensitized on malaria prevention. Apart from the auction, participants were probed on their willingness to provide non-financial support to a bio-larvicide intervention; 80 per cent stated to be willing to volunteer time to Bti application in their own locality.

Our research is methodologically more akin to another study by Mboera et al. [7], a contingent valuation study into WTP for Bti in Mvomero district, Tanzania. It relies on respondents’ yes/no answer to a single proposed amount, which is varied randomly among subsets of respondents. As such, it follows a somewhat less precise procedure than a bidding game, for its lack of iteration towards a respondent’s actual WTP. The study draws on a large sample of 962 heads of households that have at least one under-five child, located in 24 randomly selected villages.

Mean WTP was conservatively calculated at US$1.76 for a three-month period, based on a contribution rate of 73 per cent. An interesting feature of this study is that the district under scrutiny is a rice-farming area. While rice farmer status is not a sampling criterion, the presence of rice paddies plays a role in WTP, as witnessed in the focus group discussions that were held with villagers alongside the survey. Also, Matindo et al. [32], who carried out a (non-WTP) study into acceptance of Bti interventions in southern Tanzania, picked up on this contextual aspect in order to explain why their results diverge from those reported by Mboera et al.

The disparity concerns the share of respondents stating confidence that larvicides are effective in reducing malaria; 58.5 per cent (Matindo et al.) versus 91.2 per cent (Mboera et al.), despite similar shares being knowledgeable about Bti in both studies (17.8 and 19.3 per cent, respectively). Matindo et al. suggest that perceptions are influenced by the fact that Mboera et al.’s respondents “were small-scale farmers living near rice fields where water bodies are available throughout the year”, which contrasts with their own sample experiencing seasonal breeding sites [32]. We thus consider Mboera et al.’s study as most relevant point of reference when interpreting our results in Section 4.

This is not to imply that specific literature on local support for Bti application among rice farmers is non-existing. Already in the mid-2000s, Van den Berg and Knols [33] report on a pilot project where (knowledge on) bio-larviciding was integrated in farmer field schools located in Sri Lankan rice-growing communities, promoting the concept of integrated pest and vector management (IPVM). More recently, Mazigo et al. [34] conducted a mixed-methods study on the (ex-post) acceptance of a combined Bti and fertilizer intervention among 40 rice farmers in central Tanzania.

The survey part of this latter study included a question whether farmers would be willing to contribute any money to this intervention, to which all informants responded in the affirmative, but since no amounts were specified and because of the ‘packaged’ (Bti plus fertilizer) nature of the intervention, it cannot serve as a direct point of WTP comparison.

## 2. Materials and Methods

### 2.1. Study Setting

The study has been conducted in the sub-district (sector) of Ruhuha, which is located within Bugesera district of Rwanda’s Eastern Province, at 42 km from Kigali City. Rwanda is subdivided into five provinces, under which there are 30 administrative districts and 416 sectors. Sectors are in turn composed of 2148 cells and 14,837 villages. Ruhuha borders Burundi to the south and occupies an area of 54 km^2^. The elevation ranges from 1300 to 1573 m above sea level. Ruhuha is served by one health center (located in the most populous village) and hosts 140 community health workers, that is, four per village.

Since mid- 2014, a network of Community Malaria Action Teams (CMATs) has been established. A CMAT is composed of three people per village; the head of the village, a youth representative, and a community health worker. These teams help linking health professionals and the community in implementing prevention and control interventions against malaria. Setting up the CMATs was made operational by a Dutch-funded Malaria Elimination Program for Ruhuha (MEPR). The current WTP study was also carried out within the framework of this program and catered to prospective Bti application in the area.

### 2.2. Sample

The total population in the sub-district of Ruhuha equaled 17,130 individuals in 2013, spread over 4293 households and residing in five cells and 35 villages (imidugudu). Out of these, 1914 individuals were members of rice farming cooperatives operating in four different marshlands located in this sub-district, representing the 4.2 per cent of households that are engaged in (organized) rice cultivation. The following four cooperatives are included; (A) Twizamure, (B) Kopauki, (C) Inkingi y’ubuhinzi, and (D) Corinyaburiba. The cooperatives are logical entry points for WTP elicitation, as cooperatives typically offer collective purchase of inputs, such as fertilizers and insecticides, to its members, who pay for these in the form of deductions that the cooperative applies when paying the farmers’ share in rice sales.

The membership lists of the four cooperatives served as the sampling frame for this study. Participants were randomly drawn from each cooperative proportional to the size of its membership pool. With a risk of error of 5%, a confidence level of 95% and a population size of 1914 cooperative members, the minimum sample size to detect non-zero WTP was calculated at 320 rice farmers. The proportionality criterion led to 86, 30, 122, and 82 farmers being recruited from cooperatives A, B, C, and D, respectively. However, we do not claim representativeness for rice-farming households in Ruhuha as a whole, as some rice-growing families may fall outside the scope of these cooperatives. It should also be noted that our sample features a hierarchical structure, in which individual farmers are nested in cooperatives. This aspect is considered in the choice of analytical technique (see Section 2.5).

### 2.3. Data Collection

Willingness-to-pay was elicited through a specific contingent valuation technique, i.e., a bidding game procedure, which was conducted at the end of a survey in January 2015. Both were administered under field supervision of the first author by a team of ten well-trained surveyors (all Kinyarwanda-speakers). The interviewers made use of tablet devices, so that the data was immediately available in digital form (uploaded through an on-site server).

Data collected in the questionnaire include: (1) demographics, such as age, household composition, and education level; (2) engagement in rice cultivation, including the relative importance of rice in the household’s overall livelihood strategy, rice profitability, rice paddy ownership, and years of experience; (3) malaria experience, such as case incidence and frequency of mosquito bites; (4) knowledge and perceptions on larviciding, including its perceived effectiveness and safety; and (5) cooperative membership, focusing on membership duration, capital investment at start of membership, and annual contributions for running costs, fertilizers, and insecticides.

Before proceeding to the bidding game, participants were informed about the proposed intervention to tackle mosquito-borne malaria risk in their sub-district. They were told that a Bti intervention effectively reduces malaria mosquito populations and consequently malaria burden (without mention of specific expectations regarding the size of the reduction) and that it does so without causing harm to either non-target animals, human beings, or rice plants. It was explained to them that the cost of this intervention would be covered by MEPR, the foreign-funded program, but for one single season only (six months).

Therefore, rice farmers were then asked to state their maximum WTP for continuing this intervention in the future, i.e., before actually experiencing the impact of the intervention. Apart from this (ex-ante) WTP assessment, participants were also asked for their willingness to volunteer time (number of hours per week) for the manual spraying of Bti within the paddy fields belonging to their own cooperative. Finally, they were asked to reveal their personal preference for whether cooperative deductions to cover Bti cost should be lump-sum or progressive. Both the bidding procedure, as detailed below, and the questionnaire were pre-tested to maximize its success during the experimental phase. The pilot experience prompted marginal adjustments to the elicitation process and some survey questions were reformulated.

### 2.4. Bidding Game Procedure

Participants in a bidding game are presented with a sequence of (pre-fixed) bids, cued on the respondents’ level of acceptance expressed regarding previous bids to identify the individual’s switching point. This WTP technique is increasingly being used in the economic evaluation of new health care technologies. Faced with resource constraints, decision-makers in public health care systems look towards economic evaluation as a guide when selecting the most appropriate one from a range of innovative health care technologies available on the shelf. The bidding game, originally developed by environmental economists, is now well established as one possible method for such evaluation [35].

The advantage of the bidding game in which a respondent is bid up and down by the interviewer until reaching maximum WTP is that it only requires “yes or no” responses to each bid and thus has more market realism than single open-ended questions asking respondents for their maximum WTP [36]. It should be kept in mind, however, that the bidding game generates significantly higher average WTP than open-ended formats [37]. Apart from this inflation risk, eliciting WTP by means of bidding is suspected of being subject to its own specific bias; final valuations may be conditional on the starting-point chosen to initiate the bid sequence [38]. It is therefore vulnerable to ‘starting-point bias’, i.e., higher starting bids tend to produce higher accepted bids, ceteris paribus [35]. To mitigate this risk, our participants were randomly assigned to either a low or high starting bid.

Following a principle established by previous researchers, the offers following each starting bid were pre-specified by algorithms, to ensure that all subjects who had received the same initial bid were also subject to the same negotiation process [39]. Hence, the interviewers approached the participating farmers guided by a predetermined bidding algorithm with either a low or high starting bid, which was either accepted or rejected. If the participant accepted the test at that bid, the interviewer then offered a higher value, and again sought approval from him/her. In case the respondent declined the offer, then the bid was lowered and he/she was asked to reconsider the new offer. Negotiation ceased when a bid had been accepted and a higher bid rejected.

Setting the precise levels of the bids was informed by a costing exercise for Bti application in the area. Since the actual cost for all 1914 rice farmers was estimated at FRW12,162,600 (US$15,836), each farmer should contribute FRW6355 or $8.27 per cultivation season (6 months) in a lump-sum deduction scheme for full cost recovery. Alternatively, it would require a contribution of FRW1308 ($1.70) per 100m^2^ of rice paddy per season, considering that the total area of rice fields extends to 93 ha. Since cost sharing seems more realistic than full cost recovery, especially in the first few years, the (lump-sum) starting bid was conservatively fixed at FRW1500 ($1.95) as the low amount and FRW2500 ($3.26) as the high amount per cultivation season. The same way, for progressive deduction, the low and high starting bids were set at FRW300 ($0.39) and FRW500 ($0.65) per unit of land (per cultivation season), respectively.

In addition to the starting bids, the sequencing of WTP elicitation according to a lump-sum or progressive scheme, henceforth referred to as WTP_LS_ and WTP_PRO_, was also randomized. Alternating the order allows the detection of potential spillover or reference effects from one WTP elicitation to the other. Across WTP_LS_ and WTP_PRO_, a respondent remains in the same starting bid regime (high or low). This leads to four different ‘treatments’ (or algorithms), as visualized in Figure 1. Eighty respondents entered into each treatment following random assignment. Respondents were not stratified by cooperative beforehand, so that the proportionality principle with respect to cooperatives only applies at the level of the entire sample, but not treatment-wise (see Figure 1 for actual numbers of respondents per cooperative per treatment).

### 2.5. Data Analysis

The statistical analysis, carried out with IBM SPSS Statistics for Windows 27 (IBM Corp., Armonk, NY, USA), aims to shed light on the determinants of the following three variables: WTP_LS_, WTP_PRO_, and, by extension, the difference between the two. For the comparison to be informative, WTP_PRO_ first needs to be multiplied by the actual land size of a respondent in order to observe whether a lump-sum or progressive scheme would generate more resources. The difference is represented as a ratio: WTP_LS_/(WTP_PRO_*land size), so that values above (below) 1 indicate that lump-sum (progressive) generates more resources.

The explanatory variables consist of a set of individual and household characteristics, which were either directly solicited in the survey, or constructed from combining different pieces of survey information. For example, while respondents stated the (absolute) size of the land they own, their relative landholdings are calculated in terms of the distance to the mean of landholdings within the cooperative to which they belong. For the multivariate analysis variables were dummified or log-transformed if this proved to result in better model fit.

We take special interest in observing whether WTP varies systematically across the four cooperatives. However, ordinary least squares regression fails to isolate cooperative-specific effects if individual and household characteristics correlate with group membership. In order to obtain more robust estimates for the influence of being part of a given cooperative, we perform a linear mixed effects model in which cooperative membership is modelled as a random effect, while the remaining variables are assumed to be fixed effects. This approach addresses the nested structure of the data. The data are tested against the model assumptions of homoskedasticity (equal variance) and normality. The test results are reported at the bottom of the results table and the Appendix A. Two respondents were dropped from the analysis for reporting implausible zero values on their contributions to the cooperative, so that the multivariate results are based on 318 observations (bivariate results consider all 320 participants).

## 3. Results

### 3.1. Descriptive Statistics

The individual and corresponding household profiles of the sampled farmers are discussed below, supported by summary statistics in Table 1 and Table 2, respectively.

#### 3.1.1. Demographics

In terms of residence, it is interesting to note that while only marshlands with rice cultivations located within the Ruhuha sub-district were selected, 40 per cent of the sample actually resides in the neighboring sub-districts of Mareba to the north, Nyarugenge to the west and Ngeruka to the east. They are economically engaged with the Gatare, Kizanye, Nyaburiba and Kibaza marshlands that straddle the sub-district border. The mean age of participants was 44 years. Less than a quarter are young (below 35) and an almost similar share is over 54.

The generally strong participation of women in agriculture in Sub-Sahara Africa is reflected in our sample as well; close to half (45.3%) are women. A substantial share of them are widowed, a likely legacy from the genocide that Rwanda experienced in 1994. Nonetheless, over 80 per cent of participants live with a partner, most of them in a marital relationship. The education profile of participants is weak. Few have gone beyond primary schooling and 25% of respondents have not been in school at all.

#### 3.1.2. Livelihood of Farmer Households

The typical participant’s household counts six members, including three children, one of which under-five. The largest household comprises no less than 16 members. On average, households have been involved in rice farming for 12 years and their membership of a cooperative dates back eight years. From the total harvest, it is mandatory to sell 80% through the cooperative, a rule that is allegedly observed. The 20% that is kept is predominantly used for auto-consumption. A marginal share is either sold directly outside the cooperative or given for free to families and friends. Rice was the most important crop for 244 households (76.3%). For half of the remaining households, it is the second most important crop, and rice comes in third place for another third of them, after beans and cassava.

The land available for rice, expressed more conveniently in are (=0.01 ha or 100 m^2^) than in hectare, is 5.5 are for the average household, while the largest farmer owns 30 are. There are some striking differences per cooperative, however. From small to large, the mean size of lands was 3.7 (±2.5), 5.1 (±2.8), 6.0 (±4.1) and 7.8 (±3.7) for Inkingi y’ubuhinzi (C), Kopauki (B), Twizamure (A), and Corinyaburiba (D), respectively. The share capital that households have invested in their cooperative shows a rather wide range; one third has put in less than FRW5000, while for another third this exceeds FRW20,000, with the remaining third in-between.

As far as the income generated from rice cultivation is concerned, 94.4% stated that rice cultivation is either a modestly profitable or very profitable activity and a slightly smaller share (87.8%) experienced rising income from rice cultivation over the past three seasons. Income generated by rice sales rose from FRW45,040 ($59) in 2013 (second harvest) to FRW53,288 ($69) in 2014 (same harvest) on average. This upward trend could be observed in three out of the four cooperative groups. Typically, income from rice represented about 60 per cent of the household’s total income.

#### 3.1.3. Acceptability of Bti

A first result is that Bti application as such received full support among the respondents. None of them objected to the programmed intervention if no resources are required from them. They understood the causal chain that larviciding would set in motion. First, almost all rice farmers interviewed (92.8%) acknowledge that rice cultivation contributes to malaria. The majority of respondents perceived that rice fields offer a breeding site for the mosquitos that spread malaria. Moreover, 91.9% reported having experienced mosquito bites in or around rice fields often or even very often.

Seven out of ten households recount a case of malaria in their household over the past year, evidencing that the impact of malaria is widespread. The level of confidence in the effectiveness of Bti is high among respondents. All but nine (97.2%) had much confidence that larviciding will reduce the risk of contracting malaria, even though they had no prior experience applying this strategy. These results mimic fairly well those obtained by Mboera et al. [7] in a Tanzanian rice-farming area.

This high level of acceptability is reflected in the time that farmers are willing to dedicate to Bti application. Only three respondents, or less than 1%, are unwilling to volunteer any time, reportedly due to lack of time rather than for lack of support for larviciding. Half of the sampled farmers are even willing to spend more than two hours per week, as compared to 22% who restrict their labor contribution to at most an hour per week, and another 17% to a maximum of two hours.

More detailed qualitative information on perceptions of the concerned rice farming communities regarding malaria prevention, including anticipations of the Bti pilot, are documented by Ingabire et al. [40]. Focus group discussions conducted early 2015 with rice farmers, trained Bti sprayers, local leaders, CMAT members, and other community members, confirmed the above picture of high acceptability and cooperation, although some participants stressed that these are conditional on proven effectiveness and safety during the trial phase.

#### 3.1.4. Willingness-to-Pay (WTP) for Bti

After being provided with a detailed explanation of the intervention and the bidding game procedure, farmers expressed their maximum WTP for larviciding with Bti. Over the entire sample, the mean value of WTP was FRW1544 ($2.01) per season in the lump sum treatment (WTP_LS_), and FRW330 ($0.40) per are per season in the progressive treatment (WTP_PRO_). Only one respondent stated negligible WTP values. Half of the participants were willing to pay at least FRW1000 ($1.30) as a lump sum, or FRW200 ($0.26) per unit of land. While the mean values for both WTP_LS_ and WTP_PRO_ are (slightly) above the low starting bids in the game, the median values coincide in both cases with the first follow-up bid when the opening bid is rejected. The modal response in lump sum was FRW500 ($0.65), which was stated by roughly a quarter of the respondents, and FRW100 ($0.13) per are was the focal point in the progressive treatment, for which 28 per cent of the respondents settled. In the discussion section these results will be interpreted in relation to the overall cost of Bti larviciding.

When asking respondents directly whether they would prefer a lump-sum or a progressive deduction scheme, a narrow majority of 57 per cent prefers a scheme that varies with land size. While this may partly reflect stronger familiarity with progressive deductions, as contributions to collective purchases of farm inputs are typically of this nature, the size of their landholdings matters. The group that favors progressive deduction holds significantly less land on average than those who prefer a lump-sum contribution, i.e., 3.1 versus 4.1 are (*t* = 4.29; *p* < 0.05). On average, they also hold an amount of land below the mean landholding in their cooperative (−0.25 standard deviations), whereas those in favor of lump-sum hold land that is 0.33 standard deviations above the mean (*t* = 5.42; *p* < 0.05). We further explore the role of absolute and relative landholding on the actual WTP bids in the next section.

### 3.2. Factors Associated with the Willingness to Contribute

Table 3 reports on the linear mixed effects regression on (1) WTP_LS_, (2) WTP_PRO_, and (3) WTP_LS_/(WTP_PRO_*land size). Note that each of these dependent variables has been log transformed, so that the parameter values for the independent variables are not directly informative about the magnitude of change in WTP. The top part of the table considers the fixed effect variables, while the bottom part zooms in on the random effect variable of cooperative membership. Before reviewing which individual and household variables carry explanatory power, let us first consider the treatment design effects.

The sequencing of the lump-sum and progressive scenario appears to have been irrelevant to participants, ruling out contamination from WTP_LS_ on WTP_PRO_ or vice versa. However, the high starting bid significantly inflated WTP_LS_, but not WTP_PRO_. Possibly, this indicates that reference-dependent bias creeps in for higher amounts only.

#### 3.2.1. Individual and Household Factors

Overall, individual characteristics of participants do not appear to be strong predictors of WTP. Gender and age do not significantly affect WTP. Respondents who never attended school report higher WTP_PRO_, but only weakly so. When considering household composition, a higher number of (adult) family members significantly increases WTP_PRO_, but not WTP_LS_. While the expectation might be that the presence of under-five children, who are particularly vulnerable for malaria, would have a positive effect on WTP, the data do not bear this out.

In the economic domain, we find that in the case of households whose per capita income from rice is below average, WTP_LS_ is suppressed. Apart from a level effect, a trend effect of rice income is also visible. Households who witnessed a rising trend in their income from rice over the past three seasons report higher WTP_LS_. For WTP_PRO_, these income variables point in the same direction, but are insignificant. Rather, it is the household’s level of income dependence on rice that matters in this case. Higher dependency on rice strongly suppresses WTP in the progressive case, which may be explained by the fact that a malaria ‘tax’ on rice cultivation would hit them relatively hard. Consequently, lump-sum deduction would extract significantly more resources from rice-intensive households than a progressive scheme (see Column 3).

For the other variables related to rice farming, such as whether rice was the household’s most important crop, or the number of years of experience in rice cultivation, no effects were observed. Also, neither a household’s duration of cooperative membership nor its current contributions paid to the cooperative for farm inputs and running costs impact WTP significantly. Land ownership, however, proves a crucial determinant, especially when comparing WTP across the two deduction schemes.

Larger landholdings provide a strong boost to WTP_LS_, but fail to affect WTP_PRO_. The intuition is that larger farmers may feel more compelled (or able) to contribute than smaller farmers in the lump-sum case, while in the progressive case they would automatically contribute more for a given per unit deduction rate. This result should be interpreted in tandem with the parameters obtained for relative land size, as measured in terms of standard deviations above (positive value) or below (negative value) the average landholding in the cooperative.

A higher relative position in one’s cooperative strongly reduces WTP_LS_. We suspect the rationale to lie in non-linearity of the positive (absolute) land size effect on WTP_LS_ among the largest farmers in a cooperative, whose WTP likely gravitates downward towards a contribution level that is reasonable (and affordable) for farmers with more modest landholdings in a lump-sum scenario. Such a corrective tendency is less urgent under a progressive scenario, which may explain why the effect of relative land size on WTP_PRO_ is only weakly negative. The regression on the ratio of WTP_LS_ to WTP_PRO_ mimics the findings for WTP_LS_; higher (absolute) land size makes lump-sum deduction more powerful, but this advantage erodes if larger farmers are further out from the mean landholding in the cooperative.

Finally, regarding malaria experience and attitudes towards Bti, the following effects are observed. Having experienced a malaria case in the past year positively impacts WTP_LS_, unlike the reported frequency of mosquito bites, which does not motivate a higher WTP. The perceptions around Bti reveal yet an important determinant of both WTP_LS_ and WTP_PRO_, which concerns the perceived safety of Bti for other organisms. Participants who are confident that Bti does not affect the ecosystem state significantly higher WTP. Informal conversations with farmers suggest that those who entertain doubts are mainly concerned about the survival of a particular insect, to which they refer as “the farmer’s friend”, which is believed to keep crop disease at bay. Other perceptions, such as beliefs regarding the strength of the link between rice cultivation and malaria, do not show an effect on WTP, although it should be noted that perceptual variation across the sample is fairly low (see Section 3.1.3).

#### 3.2.2. The Cooperative Factor

Looking beyond individual and household factors, the bottom part of Table 3 zooms in on the random effect of cooperative membership. A first result is that both WTP_LS_ and WTP_PRO_ are significantly shaped by cooperative affiliation, i.e., alongside the impact from the predictors discussed so far. The marginal means that are estimated from the model for each cooperative provide more insight. For clearer exposition, the marginal means (and 95% confidence intervals) are back-transformed from log values into monetary values (in FRW) and compared across the four cooperatives in Figure 2a–c, corresponding to the regressions on WTP_LS_, WTP_PRO_, and the ratio, respectively. Note that these marginal means are different from actual group means, given that the former are model-based and corrected for co-variation with the fixed effect variables.

Cooperative C (Inkingi y’ubuhinzi) stands out as the most generous cooperative group in both WTP_LS_ (2a) and WTP_PRO_ (2b). Pairwise comparisons bear out that the difference is significant vis-à-vis cooperative A for WTP_LS_ and vis-à-vis A and B for WTP_PRO_. Given the fact that cooperative C has the smallest average land size among the four, the ratio of WTP_LS_ to (WTP_PRO_,* land size) is above 2, which exceeds the ratio for any of the other cooperatives (2c). Membership in cooperative B also tilts towards higher WTP_LS_ than WTP_PRO_, while the opposite is true for cooperative D (confidence interval below 1). For cooperative A the ratio hovers around 1. All pairwise comparisons on the ratio are significant.

The outlier status of cooperative C as most generous group warrants some reflection. To check the finding that higher WTP in this cooperative is not driven, at least not directly, by its particular farmer composition in terms of land size, Figure 2d reports on an auxiliary regression in which WTP is replaced by the farmers’ stated willingness to volunteer time. More specifically, it compares the estimated means obtained from running the same regression on the number of hours that respondents were reportedly willing to put into the Bti spraying campaign. Under the assumption that time investment is independent from land size, Figure 2d confirms that cooperative C displays a general tendency to be more generous. Hence, this suggests that the social norms or mindset in this group are particularly conducive for LSM.

This is not to say, however, that the particular land composition of cooperative C does not play a role. It may play an indirect part in fostering group norms. Interestingly, cooperative C embodies the largest level of land inequality among the four cooperatives, when measured according to the Palma ratio. The Palma ratio compares the land endowment of the 10 per cent largest farmers to the land endowment of the smallest 40 per cent. For cooperative C the Palma ratio equals 1.37, compared to 0.91, 0.71, and 0.76 for A, B, and D, respectively. Therefore, it is the only cooperative where the top holds more land than the bottom. Tentatively, this wide gap between small and large farmers could have bred strong solidarity norms to maintain unity in the face of stark intra-group inequalities, but a more qualitative inquiry would be required to explore this hypothesis further.

## 4. Discussion

This section relates our bidding game results to the studies reviewed in Section 1.4 and starts out by positioning the WTP levels observed. Mean WTP_LS_ for larviciding in our case has been estimated at FRW1544 (US$2.01), which is not a marginal contribution, especially when taking into account that correcting for purchasing power disparities, this equals to roughly 5 international dollars (PPP$). This is fairly in line with Mboera et al.’s [7] estimate of US$1.76 (PPP$4.40) in a rice-growing area of Tanzania, arguably the best point of comparison available. It should be noted, though, that the seasonal window used in that study was three months against six months in our study. Due to variation in site-specific seasonal weather patterns and malaria ecology in general, it is difficult to conclude whether our Rwandan estimate is higher or lower than the Tanzanian one. However, it seems safe to state that they are in the same range. Our contribution rate compares favorably, however, as it is close to 100 per cent against 73 per cent in Tanzania [34]. The US$4.20 estimate from the Kenyan Bti auction, reported in Diiro et al. [8], is substantially higher. This is likely inflated by the urban component in the Kenyan sample, despite using a revealed preference elicitation method that should put downward pressure on WTP estimates.

A pertinent question, however, is the extent to which our WTP estimates reach towards covering the actual cost of the Bti intervention in the Rwandan setting. A conservative calculation, based on median rather than mean WTP, suggests that coverage stands at around 15 per cent of the total cost, irrespective of whether deductions are lump-sum or progressive. However, this obscures substantial variation across cooperatives, if we decompose costs by cooperative. Cooperative C reaches 30–35 per cent of cost coverage, while cooperatives A and D do not exceed the 10–15 per cent range. Cooperative B takes an intermediate position with 20–25 per cent.

Since previous studies did not focus on farmers’ cooperatives, a comparison on cooperative characteristics that appear relevant in our study, such as the internal distribution of land, is precluded. However, some similarities and differences on the individual and household variables stand out. For example, in line with the results in Tanzania and Kenya [7,8], household wealth appears a more important WTP predictor than income. Wealth, in terms of land endowment, also raises WTP_LS_ in our case, whereas the positive effect from (rice-related) income is weaker. The three studies also concur on the observation that WTP is not systematically affected by a respondent’s age. On other sociodemographic characteristics, results diverge. Education increases WTP in the Tanzanian study, which we do not observe (nor in Kenya). Household size is associated with higher WTP in Rwanda and Kenya, but not in Tanzania.

Finally, the Kenyan study encounters a strong gender effect, where male respondents state significantly higher WTP than female respondents, which is not observed in Ruhuha (not reported in Tanzania).

Concerning attitudinal variables, perceived trust in the safety of bio-larvicides (not reported in Kenya), increases WTP in both Tanzania and Rwanda. Self-reported malaria incidence in the household also increases WTP in the lump-sum treatment in our study, whereas the Tanzanian study fails to find such an effect (not reported for Kenya).

## 5. Conclusions

Given that bio-larviciding has been added to Rwanda’s malaria prevention strategy, the challenge for upscaling beyond the current pilots is to embed bio-larviciding in existing community structures. While largely overlooked, farmer cooperatives hold potential as entry points for sustainable bio-larviciding. Since rice paddies serve as mosquito breeding grounds, rice farming cooperatives in particular could be part of the solution, given that they have both a direct stake and responsibility in it. Moreover, farmer cooperatives have collective financing mechanisms in place, in which contributions for spraying campaigns could be integrated.

Acceptability of Bti intervention by rice farmers in Ruhuha appears generally high, as witnessed by a stated willingness to invest (non-compensated) labor time and non-zero financial contributions. At the same time, reported WTP levels can only cover 15–25 per cent of the full intervention cost, depending on whether one believes median or mean WTP values to be the most realistic option. Even in case WTP would increase in response to proven effectiveness of larviciding, this likely leaves a substantive finance gap to be filled by other actors, such as the government, foreign donors, community members who are not involved with rice farming activities, and, perhaps, rice consumers. It seems nevertheless worthwhile to further explore the options for local resource mobilisation within cooperatives, also because of its contribution to the creation of local ownership.

Our analysis into WTP across four rice farmers’ cooperatives reveals, first, that over the entire sample a lump-sum contribution scheme (equal contributions among farmers, regardless of individual acreage) would generate similar amounts of funds as a progressive scheme, in which contributions increase proportionally with one’s land size. Larger farmers, understandably, have a preference for a lump-sum scheme. The success of resource mobilization seems more dependent on specific cooperative characteristics, however. One of the cooperatives is clearly able to tap more resources from its members for larviciding, which we tentatively attribute to stronger solidarity norms, which may in turn be related to its composition (e.g., degree of land inequality).

More detailed research into the institutional histories and dynamics of the cooperatives would be required to shed light on this, but it clearly shows that a ‘one-size-fits-all’ approach would engender inefficiencies, as it risks either pushing contribution levels beyond what is deemed reasonable in some groups, thereby jeopardizing acceptability, and/or leaving WTP potential in other groups untapped. Hence, attempts to bring on board farmer groups in LSM should leave sufficient room for bottom-up negotiations on collective contribution levels and preferred deduction modality.

## Figures and Tables

**Figure 1 ijerph-18-11575-f001:**
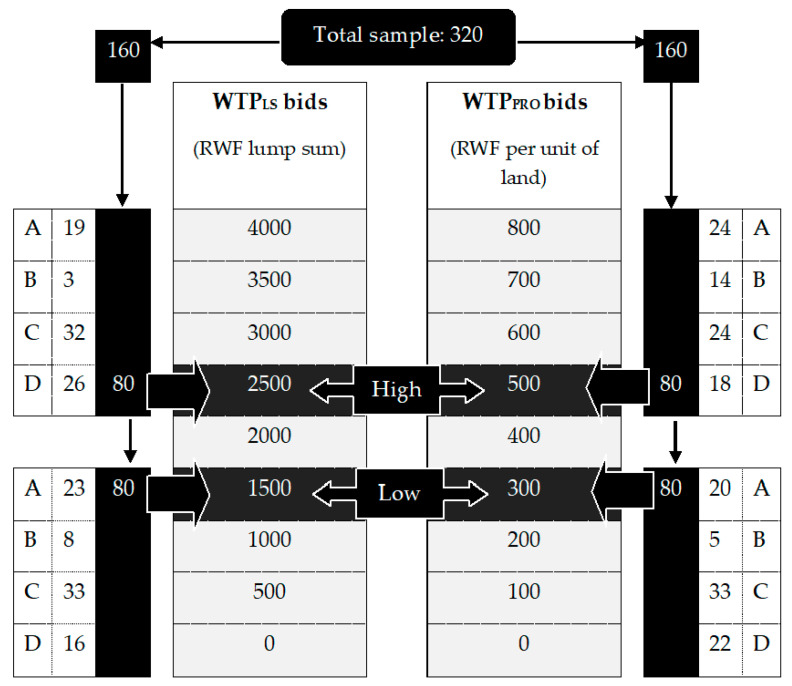
Bidding game design and sub-samples by cooperative (A–D).

**Figure 2 ijerph-18-11575-f002:**
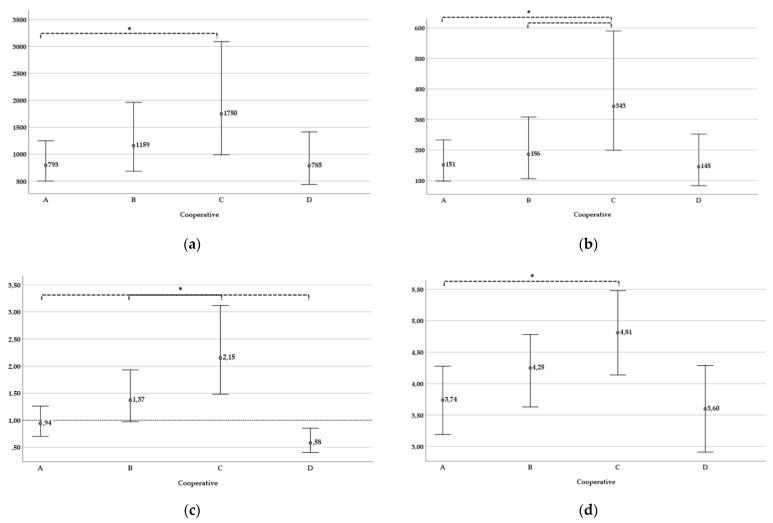
(**a**) WTP_LS_ (in FRW); estimated marginal means and 95% confidence interval, per cooperative [* difference is significant (*p* < 0.05) in pairwise comparison]; (**b**) WTP_PRO_ (in FRW per are); estimated marginal means and 95% confidence interval, per cooperative [* differences are significant (*p* < 0.05) in pairwise comparison]; (**c**) WTP_LS_/(WTP_PRO_ * are); estimated marginal means and 95% confidence interval, per cooperative [* all pairwise differences are significant (*p* < 0.05)]; (**d**) Willingness to invest time in Bti spraying (extra hours per week); estimated marginal means and 95% confidence interval, per cooperative [* difference is significant (*p* < 0.05) in pairwise comparison].

**Table 1 ijerph-18-11575-t001:** Individual profile of rice farmer respondents (*n* = 320), Ruhuha, Rwanda.

	Frequency	Percentage (%)	Cumulative (%)
**Location**			
Bihari	21	6.6	
Gatanga	22	6.9	
Gikundamvura	47	14.7	
Kindama	79	24.7	
Ruhuha	25	7.8	
Outside Ruhuha sector	126	39.4	
**Gender**			
Female	145	45.3	
**Marital status**			
Never Married	4	1.3	
Married	236	73.8	
Living together	35	10.9	
Separated/Divorced	10	3.1	
Widowed	35	10.9	
**Age group** (in years)			
<35	74	23.1	23.1
35–44	104	32.5	55.6
45–54	74	23.1	78.8
≥55	68	21.3	100.0
**Education level**			
None	78	24.4	24.4
Primary	222	69.4	93.8
Post-primary/vocational	6	1.9	95.7
Secondary school or higher	14	4.4	100.0

**Table 2 ijerph-18-11575-t002:** Household profile of rice farmer respondents (*n* = 320), Ruhuha, Rwanda.

Variable	Mean	SD	Min–Max
Number of HH members	6	2.0	1–16
Number of children below 18	3	1.8	0–10
Number of children below five	1	0.8	0–4
Number of years in rice farming	11.9	5.7	0–36
Number of years in cooperative	7.8	2.9	0–23
HH members in rice farming	2	1.0	1–9
HH members of rice cooperative	1	0.5	1–4
Rice (%) kept by the HH	20	0	20
Share (%) of rice eaten (from part kept by HH)	89.2	27.5	0–100
Share (%) of rice sold (from part kept by HH)	5.4	19.7	0–100
Size of land (in are = 100 m^2^)	5.5	3.7	1–30

**Table 3 ijerph-18-11575-t003:** Linear mixed effects regression results: WTP determinants (lump-sum, progressive, and comparison).

	(1) log(WTP_LS_)	(2) log(WTP_PRO_)	(3) log(WTP_LS_ / [WTP_PRO_*Land Size])
**Fixed effects variables**	Coeff. (st.dev) ^	Coeff. (st.dev) ^	Coeff. (st.dev) ^
Progressive treatment presented first (ref.= lump sum first)	0.000 (0.041)	0.015 (0.039)	−0.008 (0.027)
High starting bid (ref.= low starting bid)	0.087 (0.041) **	0.046 (0.039)	0.030 (0.027)
Male respondent	0.005 (0.043)	0.037 (0.041)	−0.046 (0.028)
Age respondent	−0.002 (0.002)	−0.001 (0.002)	−0.001 (0.001)
Number of HH members	0.026 (0.018)	0.035 (0.017) **	−0.009 (0.012)
Number of children under 5	0.000 (0.032)	−0.016 (0.030)	0.011 (0.021)
Number of children under 18	−0.030 (0.021)	−0.022 (0.020)	−0.011 (0.013)
Respondent did not complete primary education (dummy)	0.081 (0.051)	0.083 (0.049) *	0.025 (0.033)
Rice income per capita below sample average (dummy)	−0.097 (0.055) *	−0.085 (0.052)	−0.019 (0.036)
Share of rice income out of total HH income (%)	−0.001 (0.001)	−0.003 (0.001) ***	0.003 (0.001) ***
Rising trend of income from rice over past 3 seasons	0.086 (0.039) **	0.073 (0.057)	0.025 (0.026)
Rice farming considered unprofitable (dummy)	−0.119 (0.100)	−0.143 (0.095)	0.014 (0.065)
Rice is primary income-generating crop (dummy)	0.037 (0.059)	0.073 (0.057)	−0.063 (0.039)
Number of years in rice farming	0.003 (0.005)	−0.001 (0.005)	0.002 (0.003)
Years of membership in rice cooperative	−0.013 (0.008)	−0.011 (0.008)	−0.001 (0.006)
Land size (in are)	0.084 (0.028) ***	0.033 (0.027)	0.098 (0.018) ***
Relative land size (# of st.dev. from mean land size in own cooperative)	−0.248 (0.094) ***	−0.167 (0.090) *	−0.423 (0.062) ***
Log of share capital invested in cooperative	−0.081 (0.085)	−0.062 (0.081)	0.004 (0.056)
Log of total contributions paid to cooperative (for running costs, fertilizer, and insecticide), past year	−0.018 (0.094)	−0.043 (0.090)	−0.031 (0.061)
Any malaria case in HH in past year (dummy)	0.079 (0.045) *	0.050 (0.043)	0.048 (0.030)
Loss of any family member due to malaria (dummy)	−0.014 (0.101)	0.092 (0.096)	−0.096 (0.066)
Experience of (very) frequent mosquito bites in rice field (dummy)	−0.094 (0.097)	−0.070 (0.092)	−0.002 (0.063)
Rice field considered important breeding site for mosquito (dummy)	−0.153 (0.101)	−0.054 (0.097)	−0.122 (0.066) *
Confident that Bti will reduce mosquito population (dummy)	−0.104 (0.127)	−0.162 (0.121)	0.036 (0.083)
Trust in safety of Bti for animals (dummy)	0.086 (0.028) ***	0.097 (0.027) ***	−0.002 (0.018)
**Random effect variable**			
Membership in cooperative A, B, C or D	F = 4.44 ***	F = 5.44 ***	F = 10.128 ***
	Estimated marginal means [confidence interval]		
Cooperative A (*n* = 85)	2.899 [2.701, 3.097]	2.179 [1.990, 2.368]	−0.028 [−0.157, 0.101]
Cooperative B (*n* = 30)	3.064 [2.835, 3.293]	2.270 [2.025, 2.488]	0.136 [−0.013, 0.286]
Cooperative C (*n* = 122)	3.243 [2.995, 3.490]	2.535 [2.299, 2.771]	0.332 [0.171, 0.494]
Cooperative D (*n* = 81)	2.895 [2.641, 3.150]	2.160 [1.917, 2.402]	−0.236 [−0.402, −0.070]
Number of observations	318	318	318
Levene’s test of equal variances	F= 0.989 (*p* > 0.05)	F= 1.061 (*p* > 0.05)	F= 1.135 (*p* > 0.05)
Breusch-Pagan test for heteroskedasticity	Χ^2^ = 0.263 (*p* > 0.05)	Χ^2^ = 0.869 (*p* > 0.05)	Χ^2^ = 1.301 (*p* > 0.05)
Normality tests	see Appendix A		

^ parameter estimates based on marginal means estimates; *, **, and *** represent significance at the 10%, 5%, and 1% level, respectively.

## Data Availability

The data presented in this study are available in a publicly accessible repository and shared with the editors, with our permission to publish them.

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
