# Peer review of "Willingness to Contribute to Bio-Larviciding in the Fight against Malaria: A Contingent Valuation Study among Rice Farmers in Rwanda"

_ijerph, 2021, doi:10.3390/ijerph182111575_

Round 1
Reviewer 1 Report
In this paper, the authors explore the potential of rice farmer cooperatives in Bugesera district, Rwanda, to take up the design for campaigns to finance bio-larviciding. For this purpose, they claimed to have surveyed 320 randomly selected rice farmers in the area and elicited their willingness-to-pay (WTP) to apply Bti, a popular bio-larvicide, in their rice paddies.
The first important aspect to mention is that the topic of this study is very relevant. The authors assess a public health issue that havoc on the whole country. Their contextualization demonstrates a known origin for most of the Malaria cases in Rwanda and suggests a community approach through cooperatives to assess the problem.
Another positive aspect of this text is the high volume of data. The authors analyzed a large population and presented several aspects of the research compiled into this work. Finally, the authors performed a good set of analyses to evaluate the questionnaire data, which provided powerful insights.
Nevertheless, the text has several aspects of improving. The first one to mention is the readability. From the reviewer's standpoint, the text was hard to read in several points, and for some reason. The authors elaborated very long paragraphs, often assessing two or more different subjects. For instance, the first paragraph of the introduction is very long and discusses several topics such as the importance of mobilizing communities, disease prevention practices, LSM usage and challenges, and neighbor countries' experiences with larvicides. These topics could be easily separated into more paragraphs, which would configure a better organization. This also happens in several other locations throughout the text. Finally, given this organization, it is hard to identify the objective of the presented work in the introductory section.
On the one hand, the explanation in the introductory section is very extensive. On the other hand, the Materials and Methods section is too resumed. First of all, the text is segmented in subsections, in which most subsections are composed by an enormous paragraph alone. The lack of separation of ideas and flowcharts, itemization, and separation into paragraphs makes the text structure too compressed in these stages. This issue turns the systematization hard to notice by the reader.
There are some additional issues with this text. Although its high data value, it is poorly presented. For instance, Figures are presented split between pages (Figure 1) or low resolution (Figure 2). This problem impairs their contribution to the text. The authors should also perform some proofreading and style reviews in this text.
Given all these comments, the reviewer thinks the text requires a significant amount of work before it is suitable for acceptance.
Author Response
"Please see the attachment."

Reviewer 2 Report
Dear authors,
thank you for submitting this very interesting manuscript on an important topic. I found it well-written and easy to follow even for someone who is not deeply engaged in the social aspects of vector-borne diseases. I do have some concerns about the statistical analyses, though. Below I have made some suggestions, hoping that you'll find them useful. I think this would provide a more solid foundation for the conclusions drawn in this manuscript.
Regression model
Running a linear regression model comes with a series of well-known assumptions that need to be satisfied for the analysis to be valid. This starts with the most basic assumption of a linear relationship between response and predictors. If, for example, the WTP in relation to the number of children follows a bell curve so that farmers with very few or very many are willing to pay less than farmers with an intermediate amount of children, a linear model might detect no effect at all even though there is one. In that case, running a linear model would not be appropriate for this data, and something more advanced like a GLM would be necessary (in some cases, it might suffice to transform the data). A simple series of scatterplots of response vs. predictor would be enough to quickly assess this visually. Things like homoscedasticity are equally easy to assess. I assume that all of this has been done at some point, but as far as I can tell, none of it has found its way into the manuscript.
I might be mistaken, but the different effects observed among individuals from the different cooperatives seem to suggest that the assumption of independent sampling is violated. In any case, it seems like a good idea to account for the different solidarity norms/customs/... of the different cooperatives in the model. A (linear) mixed effects model could easily do this. A single categorical variable treated as a random effect could encode the 4 cooperatives, which would have the positive side effect of reducing the amount of dummy variables. It might be worth considering whether some of the other categorical/dummy variables should be treated as random effects as well.
Figure 2
I think using a boxplot here would be much more informative, as that includes the spread of the observed values around the median (mean could be added as an additional line if deemed necessary). This should be accompanied by statistical tests (ANOVA, Kruskal-Wallis as appropriate), to determine whether the visually observed differences between the shown groups are statistically significant. The number of samples per cooperation should also be mentioned somewhere. As a side note, I'm experiencing difficulties distinguishing the two different shades of red.
Data availability
The original data is not available, nor is there a data availability statement. The journal's instructions clearly state that "authors are encouraged to publish all observations related to the submitted manuscript as Supplementary Material" (https://www.mdpi.com/journal/ijerph/instructions#suppmaterials). I would like to express my support for that view and encourage the authors to publish (a properly anonymized version of) the original data set used for the regression analyses.
minor comments
line 410: I know the Gini coefficient is an established standard measure in economics, but as far as I know, it is not very well known in other disciplines. Suggest to add a one-liner explanation of what it does or at least add a reference for further information.
line 412: Suggest to spell out "ordinary least squares" for clarity.
line 610: Suggest to just put it into the supplement or drop it off at something like figshare. Nobody is going to be able to contact you about this in 10 years.
Author Response
"Please see the attachment."

Reviewer 3 Report
The authors report on a Willingness to Contribute to Bio-Larviciding in the Fight Against Malaria among Rice Farmers in Rwanda
Interesting and valuable information are being presented and the article would be of interest for a wide range of fields of study.
Some minor language edits are needed:
Line 25: resources not re-sources
Line 28 and 30: consistency (use same spelling for words throughout): co-operatives or cooperatives?
Line 95: Bacillus thuringiensis subsp. israelensis not Bacillus thuringiensis subsp. Israelensis
Line 184-186: add the country where the study was done
Line 443: What are the other important crops? It might not be important for your analysis but it is of interest for the reader.
Author Response
"Please see the attachment."

Round 2
Reviewer 1 Report
The authors provided an extensive review facing the first version of the manuscript. They thoroughly assessed the points presented in the last round of review. Thus, I recommend the acceptance of the present manuscript. As the authors presented a text full of markings and revisions, it was hard to check for grammar issues. Thus, I recommend reviewing it thoroughly before submitting the final version.